# Hydrophilic Natural Polylysine as Drug Nanocarrier for Preparation of Helical Delivery System

**DOI:** 10.3390/pharmaceutics14112512

**Published:** 2022-11-18

**Authors:** Bo Yu, Xiangtao Wang, Lijuan Ding, Meihua Han, Yifei Guo

**Affiliations:** 1Institute of Medicinal Plant Development, Chinese Academy of Medical Sciences & Peking Union Medical College, No. 151, Malianwa North Road, Haidian District, Beijing 100193, China; 2Key Laboratory of Bioactive Substances and Resources Utilization of Chinese Herbal Medicine, Ministry of Education, Chinese Academy of Medical Sciences & Peking Union Medical College, Beijing 100193, China; 3Key Laboratory of New Drug Discovery Based on Classic Chinese Medicine Prescription, Chinese Academy of Medical Sciences, No. 151, Malianwa North Road, Haidian District, Beijing 100193, China; 4Beijing Key Laboratory of Innovative Drug Discovery of Traditional Chinese Medicine (Natural Medicine) and Translational Medicine, No. 151, Malianwa North Road, Haidian District, Beijing 100193, China

**Keywords:** hydrophilic polypeptide, polylysine, nanoparticles, helical structure, anticancer activity

## Abstract

Polypeptide materials have clear secondary structure and biodegradability, which can be further modified and functionalized, so that they can be employed as therapeutic agents in clinical applications. PEGylation of polylysine (PEG-PLL) is a kind of safe and effective nanocarrier that is utilized for gene and drug delivery. However, PEG-PLL needs to be produced through chemical synthesis, which is expensive and difficult to obtain. We hope to simplify the nanocarrier and use hydrophilic natural polylysine (PLL) to develop a high-efficacy delivery system. To evaluate the possibility of PLL as nanocarriers, methotrexate (MTX) is selected as a model drug and PEG-PLL is utilized as control nanocarriers. The experimental results showed that PLL is an ideal polypeptide to prepare MTX-loaded PLL nanoparticles (PLL/MTX NPs). Compared with PEG-PLL as nanocarriers, PLL/MTX NPs showed higher drug-loading content (58.9%) and smaller particle sizes (113.7 nm). Moreover, the shape of PLL/MTX NPs was a unique helical nanorod. The PLL/MTX NPs had good storage stability, media stability, and sustained release effect. Animal research demonstrated that PLL/MTX NPs could improve the anti-tumor activity of MTX, the antitumor efficacy is enhanced 1.9-fold and 1.2-fold compared with MTX injection and PEG-PLL/MTX NPs, respectively. To sum up, natural polymer PLL is an ideal nano drug delivery carrier which has potential clinical applications.

## 1. Introduction

Most anticancer drugs have limited clinical application due to toxicity [1,2,3], low bioavailability [4,5,6], drug resistance [7,8,9], and other drawbacks. Encapsulating drugs with polymer materials to create nano-delivery systems is considered the potential technology to improve drug solubility, targeting, and effectiveness [10,11]. Polymeric materials utilized in the medical area need to meet the following characteristics: (1) Safety, the carrier should be non-toxic to the human body [12,13]; (2) Good biocompatibility, no immunogenicity [14]; (3) Physical, chemical, and mechanical properties must meet functional requirements [15,16,17]. With the development of macromolecules and their derivatives in the medical field, various polymers have also been produced for delivery systems [18].

Polypeptides have great prospects as a therapeutic carrier, because they present several unique properties, including secondary structure (α-helix and β-folding), good biocompatibility, excellent biodegradability, and versatile modifiability [19,20,21]. Polypeptides, such as elastin-like polypeptides [22,23,24], silk-like polypeptides [25,26], collagen-like polypeptides [27], FEK16 peptides [28], GALA peptide [29], and KALA peptide [30], are utilized as nanocarriers to deliver biological agents and small molecule drugs. Besides, zwitterionic polypeptides as an analog of polypeptides [31,32], are synthesized via chemical technology and utilized to prepare drug-loaded systems. Although these polypeptides can be utilized to transfer drugs, the complex preparation process and relatively low drug-loading capacity inhibit their further application. To overcome these drawbacks, the ideal nanocarriers based on polypeptides should present a highly active site that could interact with drugs to form a stable delivery system. Therefore, polylysine (PLL) and polyglutamic acid (PGA) are used to construct polypeptide-based nanocarriers. As a common hydrophilic nanomaterial [33,34,35], polyethylene glycol (PEG) can enhance the aqueous solubility of anticancer drugs, prolong the circulation time, reduce immunogenicity, and escape the capture of the reticuloendothelial system, which is approved by FDA and utilized as hydrophilic portion in amphiphilic copolymers to prepare nanoscale drug delivery system. Hence, PEGylated polypeptide is utilized broadly as drug nanocarriers [36,37,38].

Although PEGylated polypeptides are utilized to deliver lots of hydrophobic anticancer agents [20], several drawbacks inhibit their further clinical application. For example, these PEGylated polypeptides are synthesized via a recombinant method, including organic coupling reaction, solid-phase peptide synthesis, ring-opening polymerization, and so on [39,40,41]. To obtain the aimed block copolymers, it needs several synthetic procedures, which induce low yields and may result in batch-to-batch variation. These drawbacks seriously hinder the clinical application of amphiphilic copolymers.

Using unmodified hydrophilic peptides as drug carriers is also in line with this principle. For example, the construction of a nano-delivery system of doxorubicin encapsulated with PGA can improve the anti-tumor effect of doxorubicin and reduce its cardiotoxicity [42]. Besides, polypeptides present secondary structures, while, these effects are not shown in these delivery systems with polypeptides-based copolymers as nanocarriers [43,44], these nanoscale drug delivery systems (NDDSs) show spherical morphologies in general. It seems that secondary structures of polypeptides in block copolymers exhibit low influence on the morphologies of NDDSs. According to previous studies, it is obvious that the morphologies of drug-loaded nanoparticles (NPs) affect the therapeutic efficacy significantly, the non-spherical NPs present better therapeutic efficacy [45,46]. Therefore, it is valuable to explore the effective technology to construct drug-loaded NPs only with hydrophilic polypeptides as nanocarriers, which could present a helical structure owing to the unique secondary structure of polypeptides.

In previous studies, it is found that hydrophilic carriers, such as PEG and OEG derivatives, can encapsulate hydrophobic drugs via a suitable method to create nanoparticles [47,48]. In this study, the possibility of hydrophilic polylysine (PLL) as delivery carriers was researched, and drug-loaded NPs were prepared with the hydrophobic drug methotrexate (MTX). Then, the particle size, morphology, conformation, drug release characteristics, anti-tumor effect, and toxicity of PLL/MTX NPs were investigated in detail, demonstrating the potential of PLL as a delivery carrier.

## 2. Materials and Methods

### 2.1. Materials

Polylysine (PLL, Mn 2000-5000, Lot A2002048) was purchased from Aladdin, Shanghai, China. PEG-PLL (PEG_45_PLL_35_, Mn 7000, Lot 2129051115) was purchased from Guangzhou Tanshui Technology Co., Ltd. (Guangzhou, China). Methotrexate (MTX) was received from Shanghai Acmec Biochemical Co., Ltd. (Shanghai, China). Zirconia beads (5 mm) were purchased from Beijing Xinmei Hongxin Technology Co., Ltd. (Beijing, China). HPLC grade methanol was obtained from Fisher Scientific (Waltham, MA, USA). Other reagents and solvents were purchased at reagent grade and used without further purification.

### 2.2. Cell Lines and Animals

The 4T1 cell lines were supplied by the cell center of Peking Union Medical College. The cells were cultured in RPMI 1640 medium (HyClone, Logan City, UT, USA) supplemented with 10% FBS, 100 U mL^−1^ penicillin, and streptomycin (Gibco, St Louis, MO, USA) in a humidified 5% CO_2_ atmosphere (Sanyo, Osaka, Japan) at 37 °C.

Female BALB/c mice (6–8 weeks old, 20 ± 2 g) were obtained from Vital River Laboratory Animal Technology Co., Ltd. (Beijing, China). All animal experiments are in line with the ethical guidelines laid down by the Institute of Medicinal Plant Development (Beijing, China). The animals were raised under standard SPF conditions with free access to food and water.

### 2.3. Preparation of Drug Loaded PLL, and PEG-PLL Nanoparticles (NPs)

A 20 mL capacity vial was filled with 5 mm zirconia beads (15 g) and a magnetic stir bar. PLL (15 mg) and MTX (45 mg) were accurately weighed, placed in this vial, added with 5mL of deionized water, and stirred at 300 rpm for 4 h. After collecting from vials, the mixtures were homogenized 3 times with a high-pressure homogenizer at 1560 bar, and then the yellow PLL/MTX NPs solution was obtained.

After accurate weighing of PEG-PLL (15 mg) and MTX (45 mg), PEG-PLL/MTX NPs were obtained by adding 5 mL deionized water in a capacity vial and stirring at 300 rpm for 12 h.

### 2.4. The Diameter and Morphology of Drug-Loaded NPs

The polydispersity index (PDI), hydrodynamic diameter, and zeta potential were measured by a Malvern Zetasizer 3000 system (Malvern Instruments Ltd., Malvern, UK). The morphology of nanoparticles was observed by negative staining method under 80 KV acceleration voltage by transmission electron microscope (TEM, JEM-1400, JEOL, Tokyo, Japan). Several droplets of nano-particle solution were dropped on the carbon-coated copper mesh, air-dried at room temperature, and dyed with 2% (*w*/*v*) uranyl acetate solution.

### 2.5. Measurement of Drug-Loading Content (DLC)

The DLC of the two NPs were obtained by the following method. The freeze-dried powders of two NPs were weighed and fully dissolved in methanol. The content of methotrexate was measured using HPLC (UltiMate3000, DIONEX) on a Thermo C18 column (4.6 mm × 250 mm, 5 μm). The UV detection wavelength was 306 nm, and a calibration curve generated from methanol/Na_2_HPO_4_ solution (0.01 mol∙mL^−1^) (16/84, *v*/*v*) (y = 0.2103x − 0.1048, R^2^ = 0.9992). Each sample injection volume was 20 μL, and the injection flow rate was 0.8 mL∙min^−1^. The DLC was calculated as follows:DLC% = weight of MTX in NPs/weight of the NPs × 100%(1)

### 2.6. Circular Dichroism (CD) Spectra Analysis

The CD data were obtained by scanning at a far-UV range (190–280 nm) using a Chirascan Circular Dichroism spectrometer (J-1500, JASCO, Tokyo, Japan) with a 2 mm path-length quartz cuvette at 20 °C. The scanning speed was 500 nm per 1 min. The samples used for circular dichroism analysis were carrier material (PLL) and NPs (PLL/MTX NPs). The CD data were expressed in terms of ellipticity (θ).

### 2.7. X-ray Diffraction (XRD) Analysis

XRD of the samples (MTX, carrier materials, the physical mixture of carriers and MTX, and the freeze-dried powders of NPs) was performed using an X-ray diffractometer with graphite-filtered CuKa radiation (λ = 1.54 Å) with an X-ray diffractometer (Rigaku, Tokyo, Japan) under 40 kV and 100 mA at a scanning rate of 8 min^−1^ (2θ from 10° to 90°) at room temperature.

### 2.8. Stability Study of NPs

All the NPs were sealed at 4 °C and the samples were collected at 0, 2, 4, 6, 8, 10, 12, and 14 days respectively. The size distribution of NPs was measured three times in parallel.

The two NPs solutions were mixed with 2 × PBS (1/1, *v*/*v*), plasma (1/4, *v*/*v*), 1.8% saline solution (1/1, *v*/*v*), and 10% glucose solution (1/1, *v*/*v*) at 37 °C. The particle size of the samples was measured every 2 h within 8 h.

### 2.9. In Vitro Drug Release Profiles

Two NPs solutions and methotrexate DMF solution (1 mg/mL, 2 mL, MTX equivalent concentration) were loaded into dialysis bags (MWCO 8000~14,000 Da), which were presoaked in deionized water, respectively. These dialysis bags were put into 1 L glucose solution at 37 °C. 50 μL of internal liquid released at the predetermined time was taken and 50 μL of glucose solution was replenished. The released MTX was determined by HPLC, and the cumulative release curve was drawn according to the data.

### 2.10. Cytotoxicity Assay

4T1 cells in the logarithmic growth phase were seeded in a 96-well plate (8 × 10^3^ cells/well). After 24 h incubation, the MTX solution, PLL/MTX NPs, and PEG-PLL/MTX NPs were diluted into 0.005, 0.01, 0.05, 0.1, 0.5, 5, 10, 100, and 500 μg mL^−1^ (MTX equivalent concentration) with fresh RPMI-1640 and added into the well, the injected volume was 150 μL per well. Meanwhile, the same volume of culture media was added to the well as the blank control. Renewed the culture medium once after incubation for 48 h. 10 µL CCK-8 solution was added to each well and incubated for 1.5 h. The OD value was measured at the wavelength of 450 nm by a microplate instrument. The half inhibitory concentration (IC_50_) was determined by GraphPad Prism 5 software (the version number: 5.01). The formula for calculating the cell inhibition rate was shown in Equation (2).
Cell inhibition rate (%) = (1 − OD value of the sample groups/OD value of the blank group) × 100%(2)

### 2.11. Study of Anti-Tumor Efficacy

0.2 mL 4T1 cell suspension (1 × 10^7^ mL^−1^) was injected subcutaneously into the right axilla of BALB/c mice to induce tumor. When the tumor enlarged to an appropriate volume (150 mm^3^), the mice were randomly divided into four groups (n = 6), which were treated with 5% glucose solution (negative group), MTX injection (positive group), PLL/MTX NPs and PEG-PLL/MTX NPs (test group) via the tail vein injection, respectively. Each injection volume was 0.2 mL, and the treated dose was MTX equivalent concentration of 5 mg Kg^−1^. The drug was given every 2 days for 12 consecutive days. The volume of the tumor was recorded (length × width^2^)/2), and the weight was weighed. After being sacrificed, tumor tissue was extracted and weighed, and the formula of tumor inhibition rate (TIR) was as follows:TIR (%) = (1 − average tumor weight of treatment group/average tumor weight of negative group) × 100%(3)

### 2.12. Statistical Analysis

Comparison between groups was carried out by one-way analysis of variance software (ANOVA) (SPSS 21.0, USA). *p* < 0.05 indicated statistical significance.

## 3. Results and Discussion

### 3.1. Preparation of Methotrexate-Loaded Nanoparticles (MTX-Loaded NPs)

Grinding is a “Top-down” principle preparation method [49]. Through the shear force produced by grinding medium such as zirconium beads during stirring, the crushing force acts on hydrophobic drugs to obtain the small particles, which might be combined with hydrophilic carriers to form stable drug-loaded NPs [50]. Then, the high-pressure homogenization method was used to make the NPs smaller and more uniform [51].

To obtain uniform methotrexate-loaded nanoparticles (MTX NPs) with small diameter and good stability, these two technologies are combined to prepare MTX-loaded polylysine (PLL) NPs (PLL/MTX NPs). During the preparation process, stable PLL/MTX NPs are formed via the electrostatic interactions between the carboxyl group of methotrexate and amine groups of PLL, which is a stronger interaction than other Van der Waals force, hydrogen bonds, and hydrophobic interactions [52]. After preparing successfully, a yellow translucent solution of PLL/MTX NPs is obtained with a total yield of 76.5%. Meanwhile, PEG-PLL as the classical amphiphilic block copolymer is utilized to construct control PEG-PLL/MTX NPs with a total yield of 71.3%.

### 3.2. Characterization of NPs

PLL and PEG-PLL could be dispersed in an aqueous solution directly, the particle sizes were 880.5 and 435.6 nm, respectively, and both of them showed broad polydispersity index due to the high density of positive charges in polymer chains. After entrapping MTX, drug-loaded nanoparticles showed a compact structure owing to the electrostatic interaction between PLL and MTX, the hydrodynamic diameters of PLL/MTX and PEG-PLL/MTX NPs were 113.7 and 201.3 nm, separately (Table 1), the polydispersity index was decreased significantly (Figure 1a,c). The particle size of PEG-PLL/MTX NPs was larger than that of PLL/MTX NPs. The possible reason was that the PEG-PLL/MTX NPs presented a thick hydration layer due to PEG chains. The DLS curves of the two NPs showed a single peak, and the PDIs were less than 0.3, indicating that the two NPs exhibit good uniformity and stability in an aqueous solution. After detecting via HPLC, DLC was 47.3% and 58.9% for PEG-PLL/MTX NPs and PLL/MTX NPs separately. PLL/MTX NPs present higher DLC, because PLL shows more amine groups compared with PEG-PLL under the same concentration.

According to the TEM, PLL/MTX NPs presented rod-like morphology (Figure 1b), with an aspect ratio of 9.9: 1, and an average particle size of approximately 133.8 nm, while PEG-PLL/MTX NPs were nanosheets (Figure 1d), with an average diameter of 185.7 nm. The different morphology of these two NPs can be attributed to the different self-assembly processes of two systems. Based on the previous reports, it is clear that PLL can form α-helix through supramolecular assembly [53,54,55]. After physically encapsulating hydrophobic MTX, the morphology of PLL/MTX NPs is similar to the α-helix of polypeptides. On the other hand, α-helix is not shown in PEG-PLL/MTX NPs, which present a similar morphology with the nanocrystal of MTX. It was reported that both nanocarriers and hydrophobic drugs can affect the self-assembly process [56,57], and the final supramolecular assembly properties are influenced by several factors. The helical morphology of PLL/MTX NPs demonstrated that PLL was the key parameter, which decided the self-assemble process. Moreover, the shape of nanoparticles would affect the cellular uptake ratio and result in different anticancer activity [45,58]. According to published papers [59,60], the helical-rod shape of PLL/MTX NPs should present better activity.

### 3.3. CD Study

From the above electron microscope images, PLL/MTX NPs present a helical rod-like shape, thus PLL and PLL/MTX NPs were detected with circular dichroism to analyze their conformation furthermore [61]. The results of circular dichroism were shown in Figure 2. In general, the α-helix structure of peptides had obvious characteristics, that is, a strong positive peak appeared at 193 nm, while a negative peak appeared at 208 and 222 nm [62]. All of the characteristic peaks were shown in both PLL and PLL/MTX NPs, indicating the conformations of PLL and PLL/MTX NPs were α-helix [63]. It is reported that the secondary structure of PLL is charged coil at a low pH value or neutral condition due to the repulsive forces. While, the secondary structure can be varied by increasing the pH value or adding other negatively charged polypeptides, which can decrease the repulsive forces. In this study, the structural change can be explained by a similar phenomenon. Before adding MTX, PLL shows good water solubility and forms a charged coil owing to repulsive forces between cationic amine groups. After loading hydrophobic MTX, PLL/MTX NPs are formed via electrostatic interactions, which reduces the repulsive forces. Therefore, the secondary structure changes to a compact α-helical structure.

### 3.4. X-ray Powder Diffraction (XRD) Study

XRD could be utilized to identify the existence status of MTX in NPs. Figure 3a exhibited the XRD spectra of four samples related to PLL/MTX NPs. The XRD pattern of pure methotrexate from 0 to 90° expressed remarkable peaks approximately at 9.3°, 11.5°, 12.8°, 14.3°, 16.0°, 17.7°, 19.1°, 21.4°, 24.2°, 25.4°, 26.8°, proved that MTX has a crystal structure [64,65,66]. The strong peaks can also be detected from the physical mixture of MTX and PLL, indicating that MTX in the mixture exists in the form of crystallization. While, in the freeze-dried powder of PLL/MTX NPs, the typical diffraction peak of MTX was weakened, which indicated that the crystal structure of MTX was weakened during the preparation of PLL/MTX NPs. The data in Figure 3b demonstrated that the situation of PEG-PLL is similar to that of PLL, and the crystal structure of MTX was reduced during the manufacture of PEG-PLL/MTX NPs. These phenomena could be explained that MTX was not physically mixed with PLL or PEG-PLL, the electrostatic interactions existed in both PLL/MTX NPs and PEG-PLL/MTX NPs.

### 3.5. The Stability of NPs

The stabilities of NPs including storage stability and media stability were detected in detail. The storage stability of the PLL/MTX NPs and PEG-PLL/MTX NPs at 4 °C for 14 d was shown in Figure 4a,b. The average particle size of PLL/MTX NPs, and PEG-PLL/MTX NPs was stable at about 139.6 nm, and 223.6 nm, respectively. Compared with the initial diameter, both of these two NPs were enlarged slightly, which could be explained by Ostwald ripening [67].

Then, the media stability was investigated in different media, including PBS buffer solution, plasma, 0.9% normal saline, and 5% glucose solution. Two non-salt systems (plasma, 5% glucose solution) could stabilize the existence of nanoparticles (Figure 4c,d). The possible mechanism was that the inorganic salt system will destroy the interaction force in the NPs, so that the NPs could not exist stably.

### 3.6. Study on Drug Release Kinetics In Vitro

To simulate the release of the drug in vivo, the in vitro release curve was measured in a 5% glucose solution, and the results are shown in Figure 5. The complete release of MTX injection was shown in the initial first day, indicating the burst release of free MTX. Compared with MTX, these two NPs showed sustained release characteristics, the release procedure could be sustained for 7 day. The cumulative release rate of PEG-PLL/MTX and PLL/MTX in the initial 12 h was approximately 35%. Then, the slow-release rates were shown in the following 6 days.

Free MTX had the fastest release rate, which adhered to the zero-order equation. While the NPs exhibit sustained-release qualities due to the electrostatic interaction between the carrier and the drug. Both the two NPs released slower than free MTX. The release behaviors of the two NPs corresponded to the Higuchi equation, and the two NPs had similar release rates in each period, indicating that the two NPs had similar sustained-release characteristics.

### 3.7. Cytotoxicity Assay

The results of the CCK-8 method for the cytotoxicity of each sample and MTX injection on 4T1 cells were shown in Figure 6. The cell inhibition of the three samples showed a concentration-dependent manner, that is, the cell inhibition rate was enhanced with increasing MTX concentration. However, at the same concentration, the inhibitory effect of the two NPs on 4T1 cells was stronger than that of free MTX (*** *p* < 0.001). The IC_50_ value of free MTX, PEG-PLL/MTX NPs, and PLL/MTX NPs was 6.30, 2.03, and 1.23 μg mL^−1^, respectively. Compared with free MTX, the cytotoxicity of PEG-PLL/MTX NPs and PLL/MTX NPs was raised roughly 3.1-fold and 5.1-fold after 48 h incubation. Besides, the cytostatic effect of PLL/MTX NPs was the highest among these three samples, and the cytotoxicity was about 1.7-fold higher than that of PEG-PLL/MTX NPs (^##^ *p* < 0.01).

The possible mechanism leading to the stronger inhibitory effect of the NPs was that the NPs deliver MTX into the tumor cells through endocytosis, whereas the free MTX diffuse into the cells through passive diffusion, endocytosis had a higher transport rate than passive diffusion.

The responsible mechanism for the different cytotoxicity of the two NPs may be the distinct morphologies of the two NPs. Some investigations have indicated that the shape of the particles will have a great impact on the cell absorption behavior and anti-tumor activity [43,68]. Rod-shaped NPs show higher cytotoxicity, the larger the aspect ratios of rod-shaped NPs induce the higher the apoptosis rate [58,69]. PLL/MTX NPs possessed a distinctive helical rod-like structure, but PEG-PLL/MTX did not, it is reasonable that PLL/MTX NPs demonstrated the highest cytotoxicity.

### 3.8. Anti-Tumor Efficacy

To evaluate the anti-tumor efficacy of these two NPs in vivo, the tumor volume, tumor weight, and related tumor inhibition rate (TIR) were recorded. Although the tumor volume increased for all four groups, a different change tendency was shown (Figure 7a). Compared with the blank control group (glucose solution), MTX injection presented moderate anti-tumor efficacy, the tumor volume increased from 153.0 to 1306.3 mm^3^; the tumor volume of the two NPs groups was raised slowly, which was from 155.1 to 984.2 mm^3^ for PEG-PLL/MTX NPs and from 152.7 to 785.8 mm^3^ for PLL/MTX NPs, respectively. PLL/MTX NPs present the highest anti-tumor activity (^$$$^ *p* < 0.001, compared to glucose solution).

The TIR was calculated by the tumor weight furthermore (Figure 7b). The average tumor weight was 3.40, 2.07, 1.37, and 0.93 g for glucose solution, MTX injection, PEG-PLL/MTX NPs, and PLL/MTX NPs group separately, the tumor inhibition rate was 39.0%, 59.6% and 72.6% for MTX injection, PEG-PLL/MTX NPs and PLL/MTX NPs group correspondingly. Compared with the MTX injection group, both these two NPs groups exhibited stronger anti-tumor activity, the tumor inhibition rate was promoted 1.5-fold (*** *p* < 0.001) and 1.9-fold (*** *p* < 0.001) for PEG-PLL/MTX NPs and PLL/MTX NPs groups, respectively. Compared with PEG-PLL/MTX NPs group, the tumor inhibition rate of the PLL/MTX NPs group was promoted 1.2-fold (^#^ *p* < 0.05).

NPs were better than free MTX in anti-tumor efficacy. The possible mechanism is that the EPR effect led to the passive aggregation of NPs in the tumor tissue [70]. The stronger anti-tumor effect of PLL/MTX NPs than PEG-PLL/MTX NPs may be based on its unique helical rod shape. Some studies have reported that rod-shaped particles are more likely to aggregate at the edge of the blood vessel wall [71], and the abundant and special vascular structure at the tumor makes the rod-shaped particles have an enhanced targeting effect.

## 4. Conclusions

In this study, hydrophilic polylysine (PLL) was examined as a drug carrier, and methotrexate (MTX) as the model drug to construct a nanoscale drug delivery system. Combining the grinding method and homogenization method, stable MTX-loaded PLL nanoparticles (PLL/MTX NPs) were prepared successfully. Due to the electrostatic interactions between amine groups in PLL and carboxyl groups in MTX, PLL/MTX NPs presented high drug loading content of approximately 58.9% and a small particle size of 113.7 nm. Different from the spherical morphology in other published results, PLL/MTX NPs exhibited a helical rod-like structure. To evaluate the anti-tumor efficacy of PLL/MTX NPs, the classical amphiphilic block copolymer PEG-PLL was used to prepare normal MTX-loaded NPs (PEG-PLL/MTX NPs). Both of these MTX-loaded NPs had good stability, and similar sustained-release effects. While, compared with PEG-PLL/MTX NPs, PLL/MTX NPs showed higher anti-tumor efficacy. Cytotoxicity results demonstrated PLL/MTX NPs had a lower IC_50_ value of 1.23 μg mL^−1^, and the cell inhibition rate was enhanced 1.7-fold (vs. PEG-PLL/MTX NPs). Moreover, animal experiments also confirmed that PLL/MTX NPs could improve the anti-tumor activity of MTX. Compared with PEG-PLL/MTX, the tumor inhibition rate of PLL/MTX NPs increased by 1.2-fold. All these results suggest that PLL is an ideal nanocarrier and could be utilized to prepare nanoscale drug delivery systems with enhanced anti-tumor activity. Besides, different from traditional amphiphilic block copolymers, PLL as a natural polymer hydrophilic peptide, has lower cost and more convenient access. Based on these unique advantages, PLL as a nano-carrier to construct a nano-drug delivery systems has a potential clinical application value and is worth exploring.

## Figures and Tables

**Figure 1 pharmaceutics-14-02512-f001:**
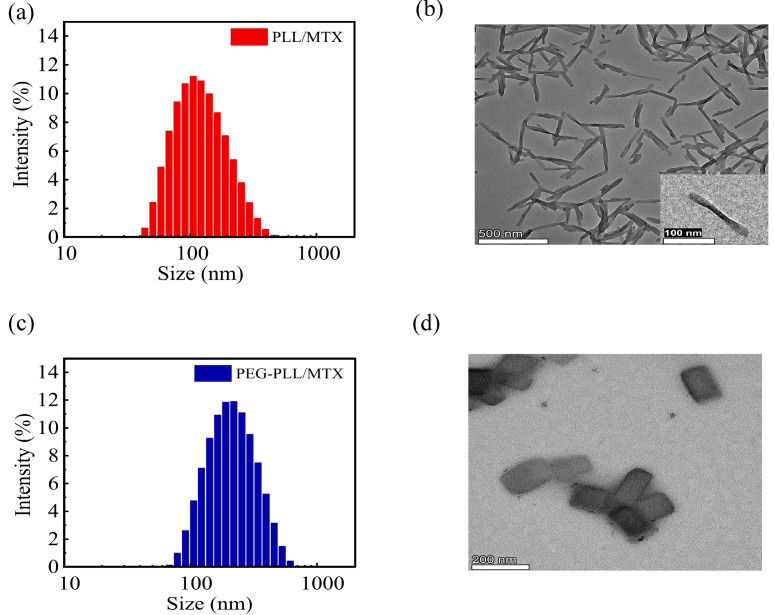
The DLC curve of PLL/MTX NPs (**a**), PEG-PLL/MTX NPs (**c**), and TEM image of PLL/MTX NPs ((**b**), Scale bar: 500 nm. insert, Scale bar: 100 nm), PEG-PLL/MTX NPs ((**d**), Scale bar: 200 nm).

**Figure 2 pharmaceutics-14-02512-f002:**
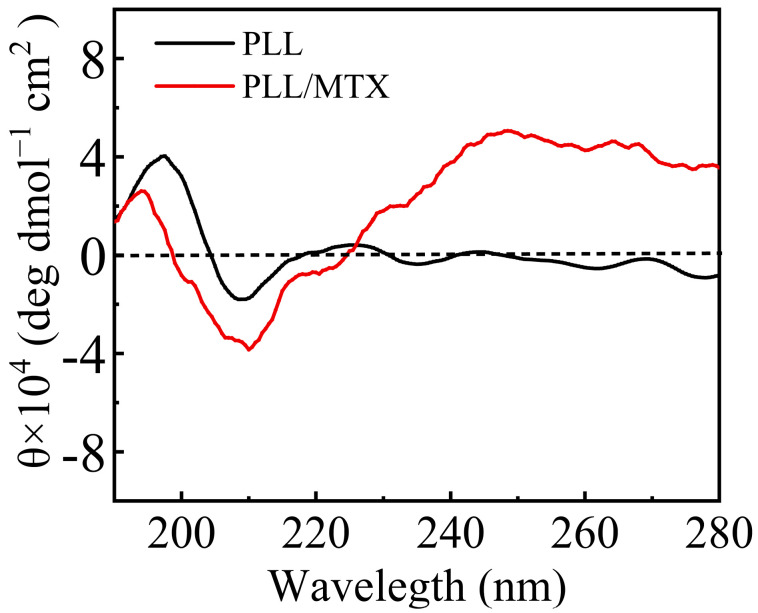
CD spectra of PLL and PLL/MTX NPs.

**Figure 3 pharmaceutics-14-02512-f003:**
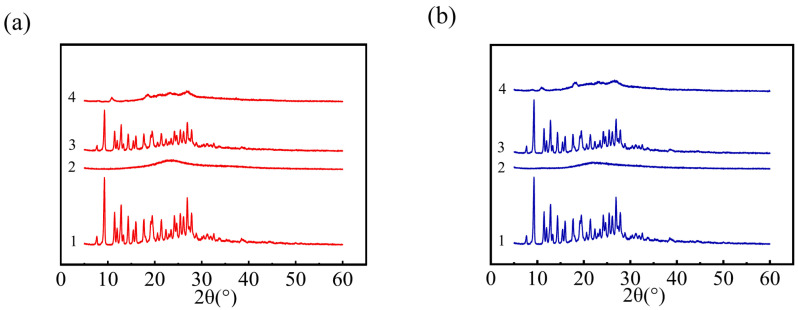
XRD patterns of PLL/MTX NPs ((**a**), 1 MTX powder, 2 PLL, 3 the physical mixture of MTX and PLL, 4 PLL/MTX NPs) and PEG-PLL/MTX NPs ((**b**), 1 MTX powder, 2 PEG-PLL, 3 the physical mixture of MTX and PEG-PLL, 4 PEG-PLL/MTX NPs).

**Figure 4 pharmaceutics-14-02512-f004:**
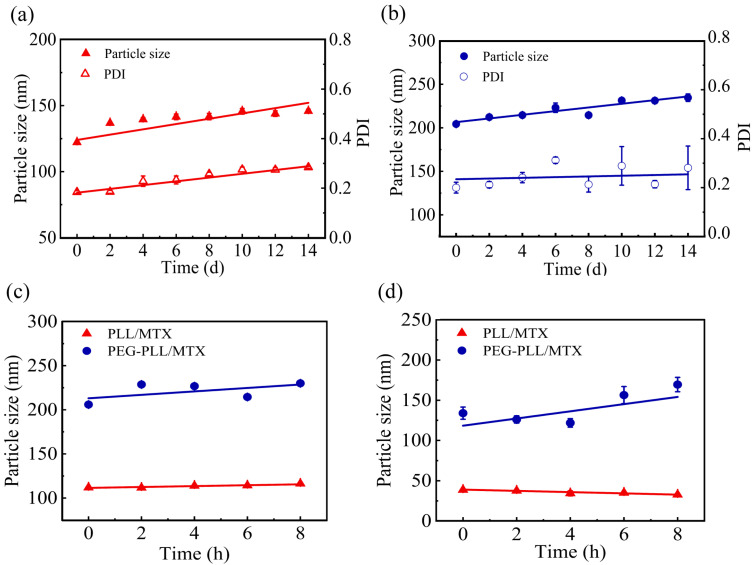
Study on the Storage stability of PLL/MTX NPs (**a**) and PEG-PLL NPs (**b**) (4 °C, 14 d); media stability of the two NPs in 5% glucose solution (**c**) and plasma (**d**) (n = 3).

**Figure 5 pharmaceutics-14-02512-f005:**
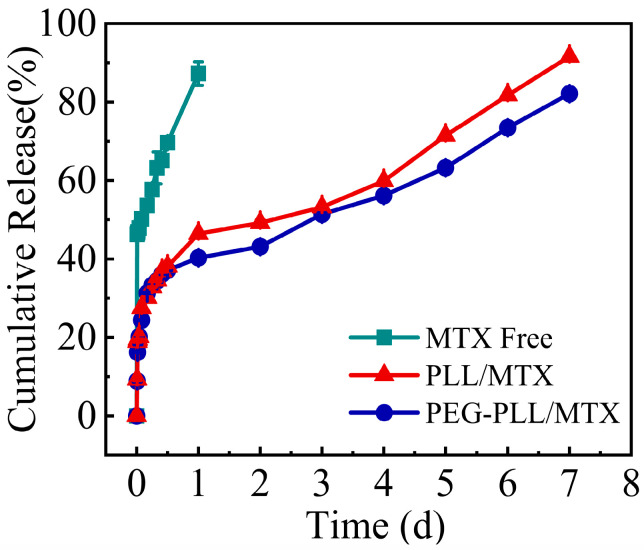
The cumulative release rate of MTX, PEG-PLL/MTX, and PLL/MTX in 5% glucose solution at 37 °C (n = 3).

**Figure 6 pharmaceutics-14-02512-f006:**
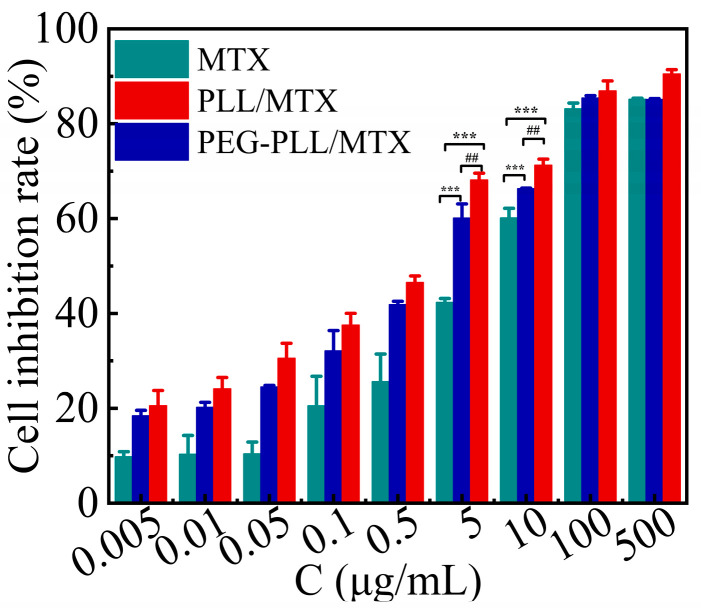
Inhibitory effect of MTX and the two NPs after 48 h incubation against 4T1 cells at 37 °C (n = 5). *** *p* < 0.001 vs. MTX injection; ^##^ *p* < 0.01 vs. PEG-PLL/MTX NPs.

**Figure 7 pharmaceutics-14-02512-f007:**
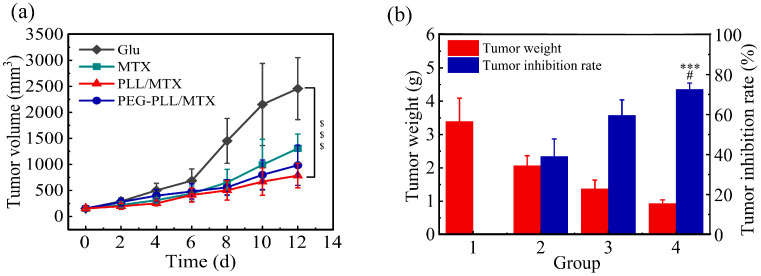
Tumor volume curves of four groups during the 12-day administration (**a**). Tumor weight and the related inhibition rate (**b**) (n = 6) (Group a: glucose solution; group b: MTX injection; group c: PEG-PLL/MTX NPs; group d: PLL/MTX NPs). ^$$$^ *p* < 0.001 vs. glucose solution; *** *p* < 0.001 vs. MTX injection; ^#^ *p* < 0.05 vs. PEG-PLL/MTX NPs.

**Table 1 pharmaceutics-14-02512-t001:** Data of the PLL/MTX NPs and PEG-PLL/MTX NPs.

Samples	DLS Results ^a^			DLC (%) ^e^
	D*_h_* (nm) ^b^	PDI ^c^	ζ (mV) ^d^	
PLL	880.5	0.69	26.6	-
PEG-PLL	435.6	0.49	21.4	-
PLL/MTX	113.7	0.23	33.3	58.9%
PEG-PLL/MTX	201.3	0.18	37.7	47.3%

^a^ Dynamic light scattering; ^b^ Hydrodynamic diameter, n = 3; ^c^ Polydispersity index, n = 3; ^d^ Zeta potential, n = 3; ^e^ Drug-loading content, UV-HPLC detected, n = 3.

## Data Availability

Not applicable.

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
