# Peer review of "Hydrophilic Natural Polylysine as Drug Nanocarrier for Preparation of Helical Delivery System"

_pharmaceutics, 2022, doi:10.3390/pharmaceutics14112512_

Round 1

Reviewer 1 Report

The present paper deals with investigation of novel polylysine (PLL) nanoparticles as a drug carrier for methotrexate. The MTX-loaded patricles were prepared and characterized in view of structure and stability in various media. The results have shown somewhat different morphology and smaller particles compared to conventional PEG-PLL NPs. On the other hand, the MTX-release kinetics in vitro was very similar to PEG-PLL, and the in vitro citotoxicity for 4T1 cells was even improved. Finally, the effects on mice tumor in vivo were also investigated, and the slight increase of tumor inhibition rate was confirmed for PLL – MTX carrier NPs compared to PEG-PLL MTX carriers.

The results of the study are interesting and significant. The concept is simple and the results are clearly presented. The idea od replacing “chemical” PEG-PLL drug carrier with pure biopolymer PLL sounds attractive, while the observed improvements of antitumor efficacy represent a bonus.

Minor questions:

-Can you provide some more information in the Introduction, on the existing research on pure polypeptides such as PLL as drug carriers?  

-The applicability of PLL carrier is limited to the hydrophobic drugs, is it? Are there some similar strategies in literature for delivery of hydrophilic drugs?

Author Response

Response to Reviewer 1 Comments Point 1: Can you provide some more information in the Introduction, on the existing research on pure polypeptides such as PLL as drug carriers? Response 1: Thank you very much for your suggestion. According to the published reports, polypeptides can be used as the carriers to deliver drugs[1,2], such as elastin-like polypeptides[3-5], silk-like polypeptides[6,7], collagen-like polypeptides[8], FEK16 peptides[9], GALA peptide[10], KALA peptide[11], and so on. These polypeptides are composed of short repeating peptide sequence, and used to deliver small molecule drugs to construct nanoscale drug delivery system. Besides, zwitterionic polypeptides as an analogue of polypeptides[12,13], are synthesized via chemical technology and utilized to prepare drug-loaded system. Although these polypeptides can be utilized to transfer drugs, the complex preparation process and relative low drug-loading capacity inhibit their further application. To overcome these drawbacks, the ideal nanocarriers based on polypeptides should present high active site which could interact with drugs to form stable delivery system. Therefore, pure polypeptides, including polylysine and polyglutamic acid are used as nanocarriers to construct drug-loaded nanoparticles. In our previous studied, polyglutamic acid is used successfully as a carrier to encapsulate doxorubicin for anticancer therapy[14]. To study the possibility of pure polylysine as nanocarriers, MTX-loaded PLL nanoparticles are prepared and its anticancer activity is evaluated in this study. The related description has been added into the introduction. (Page 2, paragraph 2) Point 2: The applicability of PLL carrier is limited to the hydrophobic drugs, is it? Are there some similar strategies in literature for delivery of hydrophilic drugs? Response 2: In this study, hydrophobic anticancer drug MTX is selected as the model to prepare nanodrug delivery system. Because the application of anticancer drugs in clinic are limited by their hydrophobicity, which results in low bioavailability and high side effects. To enhance the aqueous solubility and bioavailability, nanocarriers are explored to entrap hydrophobic anticancer drugs. Therefore, PLL is applied to encapsulate hydrophobic MTX. While, as an analogue of polypeptides, PLL can be utilized as a carrier to deliver siRNA, mRNA, etc, these biological agents present good water solubility[15,16]. Therefore, PLL should be the ideal nanocarriers, which not only can deliver hydrophobic drugs but also hydrophilic drugs.   1. Aluri, S.; Janib, S.M.; Mackay, J.A. Environmentally responsive peptides as anticancer drug carriers. Advanced Drug Delivery Reviews 2009, 61, 940-952. 2. Chambre, L.; Martín-Moldes, Z.; Parker, R.N.; Kaplan, D.L. Bioengineered elastin- and silk-biomaterials for drug and gene delivery. Advanced Drug Delivery Reviews 2020, 160, 186-198. 3. MacEwan, S.R.; Chilkoti, A. Applications of elastin-like polypeptides in drug delivery. Journal of Controlled Release 2014, 190, 314-330. 4. Chilkoti, A.; Dreher, M.R.; Meyer, D.E. Design of thermally responsive, recombinant polypeptide carriers for targeted drug delivery. Advanced Drug Delivery Reviews 2002, 54, 1093-1111. 5. Massodi, I.; Bidwell, G.L.; Raucher, D. Evaluation of cell penetrating peptides fused to elastin-like polypeptide for drug delivery. Journal of Controlled Release 2005, 108, 396-408. 6. Zhao, Z.; Li, Y.; Xie, M.-B. Silk fibroin-based nanoparticles for drug delivery. International Journal of Molecular Sciences 2015, 16, 4880-4903. 7. Frandsen, J.L.; Ghandehari, H. Recombinant protein-based polymers for advanced drug delivery. Chemical Society Reviews 2012, 41, 2696-2706. 8. An, B.; Lin, Y.-S.; Brodsky, B. Collagen interactions: Drug design and delivery. Advanced Drug Delivery Reviews 2016, 97, 69-84. 9. Jonker, A.M.; Löwik, D.W.P.M.; van Hest, J.C.M. Peptide- and protein-based hydrogels. Chemistry of Materials 2012, 24, 759-773. 10. Chen, Y.J.; Deng, Q.W.; Wang, L.; Guo, X.C.; Yang, J.Y.; Li, T.; Xu, Z.; Lee, H.C.; Zhao, Y.J. Gala peptide improves the potency of nanobody–drug conjugates by lipid-induced helix formation. Chemical Communications 2021, 57, 1434-1437. 11. Gupta, B.; Levchenko, T.S.; Torchilin, V.P. Intracellular delivery of large molecules and small particles by cell-penetrating proteins and peptides. Advanced Drug Delivery Reviews 2005, 57, 637-651. 12. Ma, G.; Lin, W.; Yuan, Z.; Wu, J.; Qian, H.; Xu, L.; Chen, S. Development of ionic strength/ph/enzyme triple-responsive zwitterionic hydrogel of the mixed l-glutamic acid and l-lysine polypeptide for site-specific drug delivery. Journal of Materials Chemistry B 2017, 5, 935-943. 13. Lin, W.; Ma, G.; Yuan, Z.; Qian, H.; Xu, L.; Sidransky, E.; Chen, S. Development of zwitterionic polypeptide nanoformulation with high doxorubicin loading content for targeted drug delivery. Langmuir 2019, 35, 1273-1283. 14. Guo, Y.; Shen, Y.; Yu, B.; Ding, L.; Meng, Z.; Wang, X.; Han, M.; Dong, Z.; Wang, X. Hydrophilic poly(glutamic acid)-based nanodrug delivery system: Structural influence and antitumor efficacy. Polymers 2022, 14. 15. Kim, S.W. Polylysine copolymers for gene delivery. Cold Spring Harbor protocols 2012, 2012, 433-438. 16. Kadlecova, Z.; Rajendra, Y.; Matasci, M.; Baldi, L.; Hacker, D.L.; Wurm, F.M.; Klok, H.-A. DNA delivery with hyperbranched polylysine: A comparative study with linear and dendritic polylysine. Journal of Controlled Release 2013, 169, 276-288.

Reviewer 2 Report

1. Size of all of the figures are very small, they should be enlarged.

2.  DLS data presentation should be improved in the results and discussion section (section 3.2, first half). DLS is a standard procedure for putting light on average hydrodynamic size and more importantly, size distribution. However, there is no discussion on size distribution of the nanoparticles before and after polymerization. The DLS column bar graph indicating size distribution should be included, relevant discussion on size distribution from the bar graph should be included in text.

3. TEM image interpretation (section3.2 second half) is also not clear. TEM analysis is the most important standard characterization for nanoparticles. The TEM images of the nanoparticle before and after polymerization looks interesting. However, authors have not provided any information on particle size and size distribution from TEM. Moreover, authors claimed an alpha-helix structure of NPs from TEM, which is irrelevant. TEM can provide information on size, size distribution and morphology, alpha-helix structure can be confirmed from CD analysis. A better discussion on TEM analysis is required. Also, HRTEM, SAED pattern and EDS experiments will help characterizing the NPs in a better way.

4. Figure 2 shows a rupture of the negative peak at 222 nm of the NPs after polymerization. This observation is interesting. However, authors have not investigated the reason for this. An appropriate explanation should be included.

5. XRD analysis (section3.4) is very poorly represented. The figure quality is very bad, which must be improved. The peaks should be clearly presented. Authors should analyze the XRD peaks using JCPDS data and interpret the results accordingly.

6. Figure 4c has an interesting observation regarding the NP stability after polymerization in glucose solution. The particle size has first increased after 2 hours then decreased after 6 hours and then again increased. Authors should investigate more on this performing some more stability experiments on different glucose concentrations. This will help shedding light on the reason behind this phenomenon.

Round 2

Reviewer 2 Report

The authors have made substantial changes in the manuscript. I am happy with most of the points. However, I am still not satisfied with the XRD figure and discussion. I would suggest to enlarge the figure and redo the experiment if necessary. MTX has multiple XRD peaks, authors have explained only two peaks, rest of the peaks are undefined. The authors must read relevant literature with MTX XRD and redo the experiment if required, they should assign crystal planes corresponding to each peaks and d-spacing and include them within the text.

In the DLS section, the figure with bar graph represents that the particles have only one size of particle i.e. particles have uniform size distribution, which is a characteristic for good nanoparticles. Authors should include this important point in text.

All of the figures are not of same size. This must be taken care of, e.g. CD figure (figure 2) looks very small compared to figure 5 and 6. Figure 2 should be enlarged like figure 5 and 6.
